# Histological, Laboratorial, and Clinical Benefits of an Optimized Maintenance Strategy of a Potential Organ Donor—A Rabbit Experimental Model

**DOI:** 10.3390/life13071439

**Published:** 2023-06-25

**Authors:** Luana Alves Tannous, Glauco Adrieno Westphal, Sergio Ossamu Ioshii, Guilherme Naves de Lima Alves, Raul Nishi Pigatto, Rafael Luiz Pinto, Katherine Athayde Teixeira de Carvalho, Júlio Cesar Francisco, Luiz César Guarita-Souza

**Affiliations:** 1School of Medicine, Pontifícia Universidade Católica do Paraná, Curitiba 80215-901, Paraná, Brazil; 2State Transplant Central, Florianópolis 88015-130, Santa Catarina, Brazil; 3Pelé Pequeno Príncipe Institute, Child and Adolescent Health Research & Pequeno Príncipe Faculty, Curitiba 80250-060, Paraná, Brazil; 4Positivo School of Medicine, Curitiba 81280-330, Paraná, Brazil

**Keywords:** clinical benefits, maintenance strategy, organ donor, experimental model

## Abstract

Introduction: Most transplanted organs are obtained from brain-dead donors. Inflammation results in a higher rate of rejection. **Objectives:** The objective of this animal model of brain death (BD) was to evaluate the effect of the progressive institution of volume expansion, norepinephrine, and combined hormone therapy on clinical, laboratory, and histological aspects. **Methods:** Twenty rabbits were divided: A (control), B (induction of BD + infusion of crystalloid), C (BD + infusion of crystalloid and noradrenaline (NA)), and D (BD + infusion of crystalloid + vasopressin + levothyroxine + methylprednisolone + NA). The animals were monitored for four hours with consecutives analysis of vital signs and blood samples. The organs were evaluated by a pathologist. **Results:** In Group D, we observed fewer number and lesser volume of infusions (*p* = 0.032/0.014) when compared with groups B and C. Mean arterial pressure levels were higher in group D when compared with group B (*p* = 0.008). Group D had better glycemic control when compared with group C (*p* = 0.016). Sodium values were elevated in group B in relation to groups C and D (*p* = 0.021). In Group D, the organ perfusion was better. **Conclusion:** The optimized strategy of management of BD animals is associated with better hemodynamic, glycemic, and natremia control, besides reducing early signs of ischemia.

## 1. Introduction

Brain death (BD) is defined as complete and irreversible damage of brain functions, characterized by apperceptive coma, absence of brainstem reflexes, and apnea [1]. BD represents the final process of an aggression generated by several mechanisms, such as ischemia and biochemical and cellular degeneration caused by initial brain injury and intracranial hypertension [2]. In the context of transplantation, the identification and diagnosis of BD is of significant importance because 73–80% of the organs are obtained from brain-dead donors [3,4]. The success of donations depends on the early detection of potential donors with BD and referral to trained professionals to carry out the correct evaluation and management of donors [2].

The succession of pathophysiological phenomena in organ donors might involve BD in the pre-transplant period, decreasing the success rate of transplantation [5]. It involves a cascade of events that follow herniation and cerebral ischemia, composed of loss of central neuro–hormonal regulatory control, hypothalamic–pituitary dysfunction, and transient increase in sympathetic activity record, resulting in the hypoxia of tissues, amplification of inflammatory response, and dysfunction of organs and tissues intended for transplant [6,7,8,9]. In addition to the increase in cytokines, deficiency of vasopressin, thyroid hormones, and cortisol contributes to increased inflammatory response [5,10,11,12].

A proper understanding of the pathophysiology of BD and the organic impairments related to it are necessary to adequately treat the inherent hemodynamic instability and also ensure the preservation of organs and reduce the delay in time for transplantation [2]. BD culminates in a cascade of serious events that leads to cerebral edema and, consequently, reduced cerebral blood flow, and reduced volume of circulating cerebrospinal fluid (CSF). This sequence causes intracranial hypertension that, if persistent above 20 mm Hg for more than 20 min, can be deleterious and cause irreversible ischemic injury [4].

Circulatory shock is a consequence of impairment of, at least, one of the following components: preload, contractility, or afterload. An impressive number of organs of potential donors are lost because of the inappropriate clinical management of the patients with BD [2]. The hemodynamic instability resulting from the lack of multiple hormones and the rise in the pro-inflammatory activity could explain the worse prognosis of patients that receive organs from brain-dead donors than that of recipients receiving organs from live patients [11,13,14,15,16,17].

Clinical studies have already demonstrated that loss of sympathetic adrenergic regulation is a moment of greatest hemodynamic instability [4]. The moment preceding the transtentorial herniation is accompanied by a significant increase in ICP and accompanied by Cushing’s triad (bradycardia, hypertension, and irregular breathing) as a final effort to ensure cerebral perfusion [4]. The adrenergic storm (the apex of hemodynamic instability) can lead to ischemia and myocardial dysfunction, which may trigger—along with regional or systemic ischemia—a potent inflammatory response, incurring severe hemodynamic instability [2].

The etiology of hypotension in potential donors is multifactorial, resulting from the loss of sympathetic tonus plus myocardial depression and associated with endocrinal and metabolic disorders [4]. Failure in management of hemodynamic disturbances leads to inadequate tissue perfusion, which is the reason why several organs are not retrieved and transplanted [4]. Reversible myocardial dysfunction in patients with severe brain injury is described as a neurogenic myocardial stunning and, when treated aggressively, can be reversed in time, increasing the number of previously rejected organs [17].

There is a significant disparity between the number of patients requiring organ transplantation and the availability of organs for donation [18]. The annual number of transplants is only 10% of the global need [19], and the low supply of organs is associated with underreporting of BD, lack of family consent, and failure in the clinical management of potential organ donors (PDs) [4,20,21,22,23].The failure of a standard maintenance practice is associated with the loss of PDs due to cardiac arrest and decrease in quality of the organs to be transplanted [14,24,25,26].

The standardization of some established physiological targets can result in a greater number of organs transplanted per donor, better graft function, and less loss of PD due to hemodynamic instability [3,6,27]. Since the beginning of the autonomic storm, it is suggested to treat the hemodynamic instability and infections, prevent diabetes insipidus, and control glucose level [1,28]. In addition, the benefits of protective mechanical ventilation, volume expansion, and vasopressin, thyroid hormone, and corticosteroid replacements have been demonstrated [12,29,30,31].

The hemodynamic instability is responsible for the loss of approximately 20% of organs from potential brain-dead donors, especially in marginal donors, in whom there is a prevalence of comorbidities associated with aging and other previous organ dysfunctions [17]. There are two existing pathophysiological mechanisms to explain the hemodynamic instability in patients with BD: the expressive increase in intracranial pressure, requiring an adequate vascular and sympathetic compensatory response in attempt to maintain the cerebral perfusion pressure (CPP) and, subsequently, the loss of vascular tone due to the abrupt drop in sympathetic tone, which may be aggravated by associated neuroendocrine disorders [17].

Hemodynamic resuscitation with fluids, for an example, can reduce hemodynamic collapse and increase the number of organs transplanted per donor, achieving adequate euvolemia, cardiac output, perfusion pressure gradients, and sufficient blood flow for organ preservation with minimal need for vasoactive drugs [24].

The wasted organs from potential donors is a problem that originates from several causes, such as refusal of donation by family members (mitigated by awareness campaigns), underreporting of BD cases and incorrect contraindications for donations (reduced by research and training healthcare teams, especially physicians) and, finally, the loss of potential donors due to cardiac arrest (potentially reduced through the use of manageable protocols, especially in the first 24 h when most losses occur [24]).

The management of patients with BD is extremely complex and requires a qualified multidisciplinary team, whose performance could be optimized with clear and well-defined management goals, aiming a significant increase in the number of donations and a reduction in the number of denials by family members [24]. Unfortunately, the focus on optimizing the conversion of potential donors into effective donors has been mainly centered to reduce the refusal of family members and not on the clinical stability of potential donors. The clinical maintenance of potential donors should be personalized in intensive care units, with different approaches for each patient directed by organs that will potentially be donated [17].

The intensivists responsible for the care of potential donors should aim to increase the availability and functionality of the organs in the long term and the use of guidelines can help in this complex process, serving as alert tools for the use of the best available evidence [2].

In this context, our main hypothesis is that the use of hemodynamic care and combined hormone therapy suggested in specialized guidelines on the subject may result in better clinical, laboratory, and histological results. Therefore, the objective of this animal model of BD was to evaluate the effect of the progressive implementation of volume expansion, norepinephrine, and combined hormone therapy on clinical, laboratory, and histological aspects.

## 2. Materials and Methods

### 2.1. Animals

Twenty male New Zealand rabbits weighing between 3.0 and 4.0 kg and aged between 4 and 5 months were included in the experiment. All procedures were carried out in accordance with the ethical principles for animal experimentation (11,794 Law of 8 October 2008) and after approval by the Ethics Committee on the Use of Animals (CEUA-project registry-01153).

The rabbits were transferred overland to the lab where operative technique and experimental surgery were performed, approximately 12 h before the start of the test. The rabbits were retained in their individual cages and maintained at a night/day cycle of 12 h, in an environment with noise control, at a temperature of 22 °C, and humidity of 60%. The rabbits were fed a standard diet and provided water *ad libitum* until 12 h before anesthesia. During the experiment, the rabbits were handled carefully to avoid an increase in physical activity or metabolic stress.

### 2.2. Treatments

The 20 animals were randomly divided into the following four groups of five rabbits each:Group A (control)—anesthesia + oro–tracheal intubation/ventilation + craniotomy + bolus infusion of 0.9% sodium chloride in case of hypotension, without exceeding 40 mL/kg;Group B (BD and volume replacement)—anesthesia + oro–tracheal intubation/ventilation + craniotomy + BD + bolus infusion of 0.9% sodium chloride in case of hypotension, without exceeding 40 mL/kg;Group C (BD and basic strategy)—anesthesia + oro–tracheal intubation/ventilation + craniotomy + BD + bolus infusion of 0.9% sodium chloride in case of hypotension, without exceeding 40 mL/kg + NA 0.05 − 2 mcg/kg/min in case of hemodynamic instability;Group D (BD and optimized strategy)—anesthesia + oro–tracheal intubation/ventilation + craniotomy + BD + induction of hemodynamic support optimized with bolus infusion of 0.9% sodium chloride without exceeding 40 mL/kg + continuous infusion of 0.04 U/h vasopressin + 15 mg/kg methylprednisolone + 4 mg/kg levothyroxine + 0.05 − 2 mcg/kg/min NA in case of hemodynamic instability.

### 2.3. Anesthesia and Induction of Brain Death

Anesthesia was induced with ketamine 35 mg/kg and 5 mg/kg xylazine, followed by oro–tracheal intubation with video laryngoscope and mechanical ventilation [32]. Analgesia with intravenous dipyrone was carried out for a longer duration than local anesthesia with lidocaine. All the rabbits were ventilated under controlled pressure with peak inspiratory pressure of 20 cm H_2_O, fraction of inspired oxygen (FIO_2_) of 100%, respiratory frequency of 50 cycles/min, and positive end-expiratory pressure (PEEP) of 8 cm H_2_O.

Two peripheral veins in the lower limbs were cannulated and the auricular artery was punctured for continuous monitoring of blood pressure.

All the rabbits were subjected to trepanation surgery. Brain death was induced in Groups B, C, and D, with the insertion of a catheter Fogarty^®^ 4 F (Edwards Lifescience LLC, Irvine, CA, USA) through the hole of craniotomy. Brain death was induced by rapid inflation of the catheter with 3 mL of air, and it was confirmed by observation of hypertensive peak, absence of corneal reflex, and bilateral fixed mydriasis [33].

After the induction of BD, anesthesia was suspended in Groups B, C, and D. Group A rabbits received extra doses of anesthetics and analgesics throughout the time of evaluation. 

To avoid hypothermia, the rabbits were heated with a thermal mattress and via hot compresses on the paws and ears. All the rabbits received infusions of medications according to the corresponding dose for each group. 

The rabbits were kept under clinical observation to monitor the mean arterial pressure (MAP), heart rate (HR), oxygen saturation (Sat O_2_), and rectal temperature for 4 h. After this period, euthanasia was performed with anesthetic overdose (intravenous sodium thiopental, at a dose of 150 mg/kg). The organs were removed and stored in formalin for the subsequent histological analyses.

### 2.4. Laboratory and Histological Analyses

Blood samples were collected for laboratory analysis at the time of anesthetic induction and at the end of the experiment, with four hours of observation. Lactate, creatin phosphokinase (CPK), potassium, sodium, glucose, hemoglobin levels, glutamic–oxaloacetic transaminase (TGO), glutamate–pyruvate transaminase (TGP), and amylase were analysed. The laboratory analysis was performed by a veterinary diagnostic center (Bionostic, Curitiba-PR, Brazil). 

At the end of the experiment, the organs were removed, photographed, and fixed with phosphate buffered 10% formaldehyde. After histological processing in paraffin, 4 µm histological sections of the liver, kidney, and heart were stained using hematoxylin–eosin and digitalized whole slide images (WSI) were obtained with an Axio Scan Z1 (Zeiss, Oberkochen, Germany). The WSI examination was performed by a blinded pathologist using ZEN Blue Edition Software 2.3 (Zeiss, Oberkochen, Germany) and Image-Pro Plus (Media Cybernetics, MD, USA). To quantify specific changes in the regions of interest in the WSI, after segmentation of the images, different color masks were applied on each slide to identify the desired variable. The area was calculated in µ^2^ and cell size or diameter of structures were measured linearly. In the kidneys, we evaluated the glomerular space and size, the cell size, and the smallest diameter of tubules. In the heart, the space between the muscle fibers to quantify interstitial oedema and the average size of the vessels to quantify congestion were evaluated. In the liver, we measured the degree of microsteatosis and the average size of hepatic sinusoid. Ten fields in each slide were evaluated to obtain the average of the variables per animal. 

### 2.5. Statistical Analysis

The results of the analysis of vital data and laboratory tests are described in line graphs and averages, histological variables, and representative average charts with a confidence interval of 95%. For comparison between groups, one-way analysis of variance was used with pos hoc Scheffe test for variables that presented normality and homogeneity of variances. The parameters that did not meet these criteria were analyzed using Kruskal–Wallis non-parametric test. Statistical analysis was performed using IBM SPSS Statistics software version 22.0, and the significance level was fixed at 5% (*p* < 0.05).

## 3. Results

A rabbit from Group B suffered cardiac arrest due to hemodynamic instability at 3 h of observation and the corresponding data were excluded from the analysis. 

The analysis of vital data showed that the body temperature of the rabbits from the three groups subjected to BD (B, C, and D) was significantly higher than that of Group A (*p* = 0.021; Figure 1A). There was no significant difference in the HR and Sat O_2_ between the groups (Figure 1B,C). 

In Group D, the MAP was higher than that of the other groups (*p* = 0.021), and the rabbits of this group did not require the infusion of NA for hemodynamic control (Figure 1D). Fewer fluid infusions were required with lesser volume for hemodynamic control in Group D. Group B required three times more fluids (*p* = 0.006) and 3.4 times more infusions that those for Group A (*p* = 0.007). However, in Group C, the amount of infusion was 2.3 times higher (*p* = 0.042) and the number of infusions was 3.6 times higher (*p* = 0.004) than those for Group A. Group D received 2.2 times more infusions than Group A (*p* = 0.108) and 1.6 times more fluids than Group A (*p* = 0.273).

The laboratory tests revealed an increase in the average level of sodium at points 0 and 3 in Group B (*p* = 0.028). There was also an increase in glucose level variation between those points in Group C compared with that in the other groups (*p* = 0.026). The lactate, potassium, hemoglobin levels, TGO, TGP, CPK, and amylase showed no differences between the groups (Figure 2).

There was no difference in the histological analysis of the kidneys, heart, and liver (Figure 3). There was a significant difference in the average size of renal tubular cells between the four groups (*p* = 0.001). The renal tubular cell size of Group D was less than that of Groups B and C (*p* = 0.003 and *p* < 0.001, respectively). However, there was no difference in the renal tubular cell size of Group D in relation to that of Group A (*p* = 0.088). The average size of renal tubular cells of Group C was higher than that of group A (*p* = 0.004). In relation to the average size of the heart vessels, there was no difference between the four groups (*p* = 0.022). Group A presented smaller average size in relation to that of Groups B (*p* = 0.003) and C (*p* = 0.036), whereas groups A and D presented a similar average size (*p* = 0.106). With respect to average hepatic steatosis, there were differences between the four groups (*p* = 0.017). Although, in the paired analysis, there was no significant difference between Groups B, C, and D, we observed that Group D presented the lowest average steatosis among them (562.6 µm^2^), being the closest to the group average (208.0 µm^2^). Figure 4, Figure 5 and Figure 6 illustrate the histological changes observed. There were no differences in the other histological parameters analyzed.

## 4. Discussion

In usual clinical practice, brain-dead patients are validated as potential donors based on the medical history, vital data over time, and laboratory tests that can demonstrate target organ damage [2,34,35]. In the present study, we assessed three different potential therapeutic donor care strategies (groups B, C, and D), commonly used in clinical practice, and verified the effect on parameters routinely used for the acceptance of an organ, which reflects the external validity.

In relation to the vital data, all the groups with induced BD presented a peak in BP soon after the procedure. This phenomenon is well known and is due to the increased level of norepinephrine, epinephrine, and dopamine immediately after BD, increasing the MAP transiently [6,12,13,33]. After this period, the rabbits developed hypotension, which was treated as per the protocol of each group. Group B, which did not receive any vasopressor, exhibited progressive hemodynamic deterioration, culminating in cardiac arrest in one of the rabbits. The rabbits from Group D were hemodynamically stable over time compared with that of rabbits in Group C as they presented a high MAP (Figure 1D), reduced need for fluids, and few number interventions to maintain stability. The HR of this group was relatively low (Figure 1B), indicating that these rabbits remained more hemodynamically compensated. These findings can be explained by the rapid depletion of vasopressin level in addition to depleting cortisol and thyroid hormone levels [12,31,36]. However, in Group D, the replacement of vasopressin, corticosteroid, and thyroid hormones, although controversial [14], seems to have facilitated the hemodynamic control [6,13,37]. Although we did not observe a significant difference, the data showed that the Sat O_2_ of Group D was superior to that of Groups B and C (Figure 1C), a condition that can be explained by better perfusion and reduced need for volume infusion, avoiding congestion and improving gas exchange [6]. In addition, there are prior reports of corticosteroid usage to improve oxygenation in potential lung donors [38,39,40,41]; however, the current evidence does not support this use. The lack of control of fluid responsiveness parameters is a limitation of our study [42]. The temperature increase observed in Groups B, C, and D (Figure 1A) was probably caused by the lesion-induced loss of brain self-regulation [41,43,44]. In most cases, hypothermia has been observed, but in this study, there was active heating of the environment of animals in order to prevent arrhythmia and hemodynamic decontrol [1,14,45,46], and hyperthermia has been reported. The control group, with preserved brain self-regulation, maintained stable temperature despite external heating. Diuresis would be an important clinical parameter to be measured in PD. Not having performed this measurement is a limitation of our experiment. The hypernatremia post BD can be explained by the following mechanisms: depletion of vasopressin, induction of diabetes insipidus and dehydration, and the use of hyperosmolar solutions [12,28]. This finding was consistent with the findings of the present study in Groups B and C (Figure 2A). Group D, that received vasopressin as treatment, did not present hypernatremia, which contributed to increased hemodynamic stability among the rabbits of the group, as described previously [36]. With respect to the pathophysiology of BD, there was a reduction in the level of insulin, which corresponded with the critically ill condition of the patient, predisposing to hyperglycemia [6,47]. There are reports of association between blood sugar control and the increase in the number of donated organs, in addition to the improvement of quality of renal grafts [47,48]. Regarding the changes in the blood glucose level, there was no significant data except in Group C (Figure 2B). These animals received NA, which can cause hyperglycemia [49,50]. Hyperglycemia was not observed in Group D, in which vasopressin was used as a vasopressor of choice. Although the data did not demonstrate statistical significance, the lactate level showed a tendency to increase in Group B compared to other groups (Figure 2C). The change was expected because these animals did not receive vasopressor treatment and developed more severe hypotension (Figure 1D) [51]. The histological parameters evaluated demonstrated hypoperfusion and early ischemia. Ischemia was based on sinusoid congestion, hypotrophy of pericentral hepatocytes in zone 3, and also on the presence of steatotic cytoplasmic microvacuolation.

When we evaluated the average tubular cell size, average cardiac vessel size, and average hepatic steatosis, Group D was the closest to Group A among the intervention groups, generating statistical significance. This finding is consistent with the better clinical stability of animals of this group, with potential repercussions on perfusion and inflammation of the kidneys, liver, and heart.

The fact that histological changes can be detected without significant change in laboratory tests commonly used for validation of organs for transplant is quite relevant. This fact suggests that tissue injuries could be detected even before the biochemical changes in the early phase of BD. 

In the present study, we focused on changes in the early hours after BD and therefore some results did not show differences between the groups. Further studies with extended observation time can provide significant insights into additional parameters. It is important to highlight that this study was performed in a small number of participants. As an experimental study, it does not intend to be conclusive but to put forward a hypothesis to identify biological variables that are difficult to systematically measure in humans. Our data demonstrated that the optimized strategy of clinical management (infusion of crystalloid + vasopressin + levothyroxine + methylprednisolone + NA, if hypotension present) increased hemodynamic stability, improved the control of sodium and glucose levels, and decreased acute ischemic signs in the heart, kidneys, and liver.

## Figures and Tables

**Figure 1 life-13-01439-f001:**
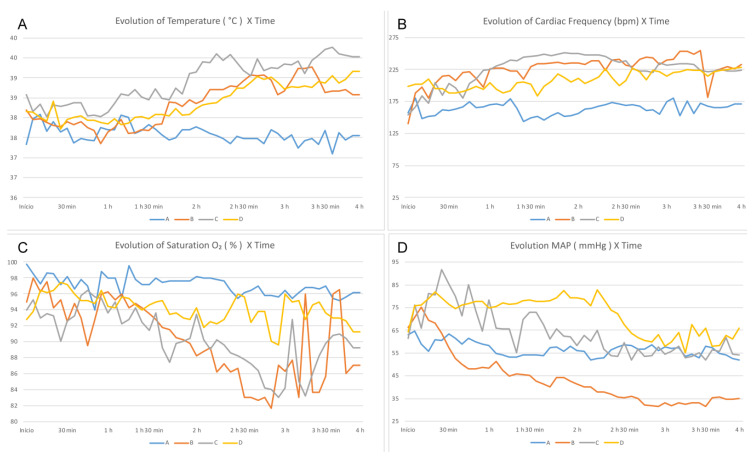
Evaluation of the vital data over time—comparison between groups. (**A**) Temperature, (**B**) HR, (**C**) Sat O_2_, and (**D**) MAP.

**Figure 2 life-13-01439-f002:**
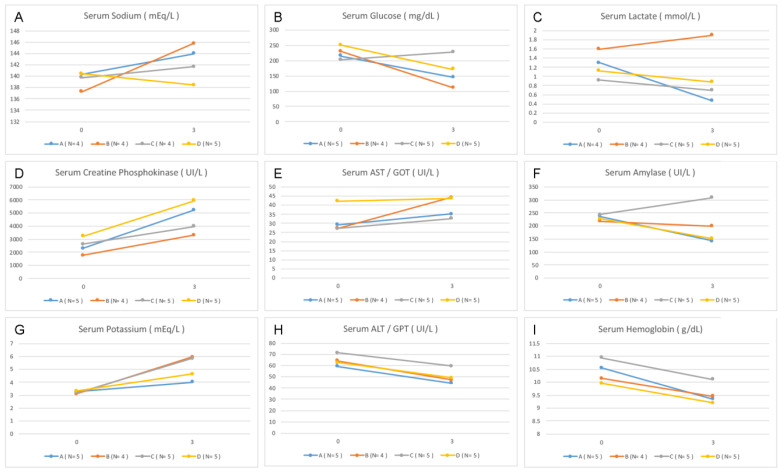
Variation in the laboratory data between the beginning and the end of the experiment. Point 0—anesthetic induction. Point 3—end of experiment (four hours of observation). (**A**,**B**) Dextrose, (**C**,**D**) CPK, (**E**) TGO, (**F**) amylase, (**G**) potassium, (**H**) TGP, and (**I**) hemoglobin.

**Figure 3 life-13-01439-f003:**
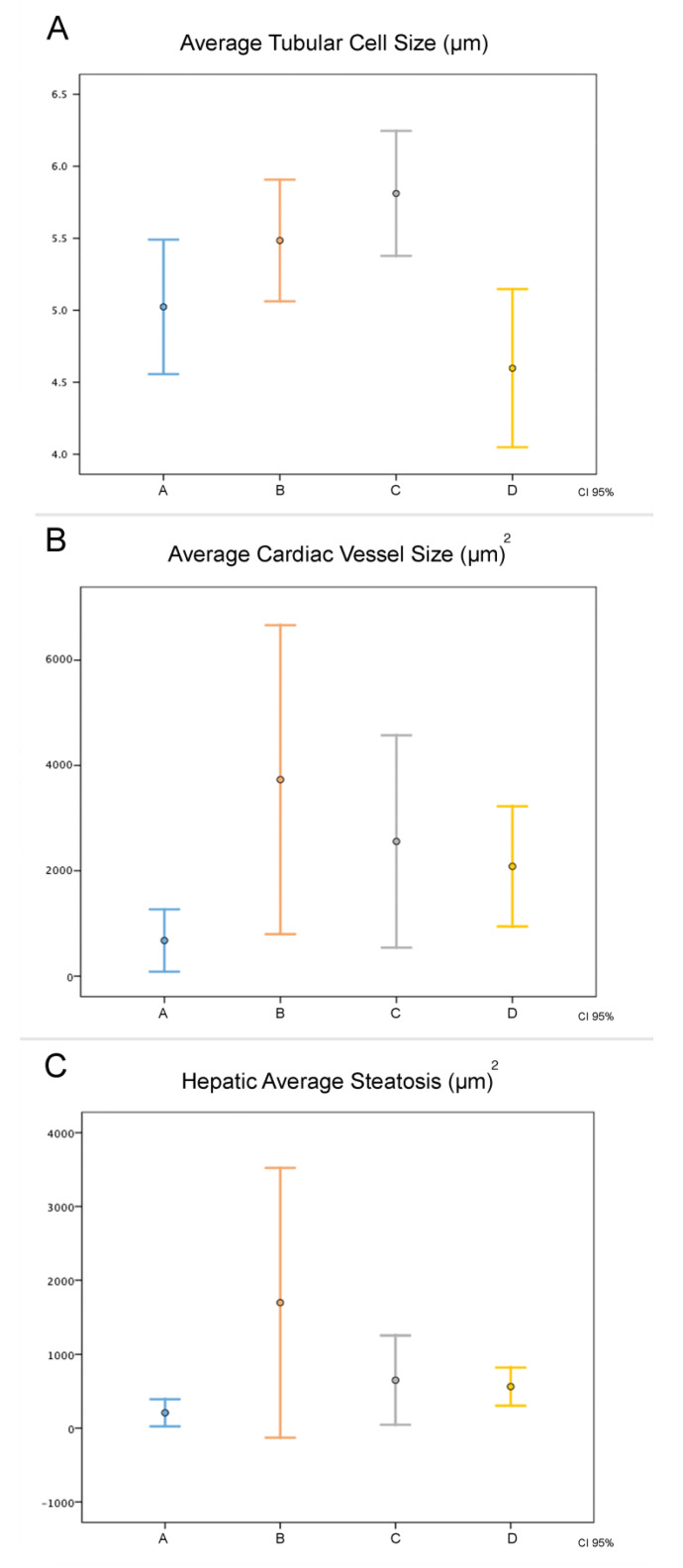
Comparison of histological data between the groups. In abscissa, the groups studied, with the results expressed by the mean (center point) and their 95% confidence interval. (**A**) The average size of renal tubular cells, (**B**) average size of the heart vessels, and (**C**) average steatosis.

**Figure 4 life-13-01439-f004:**
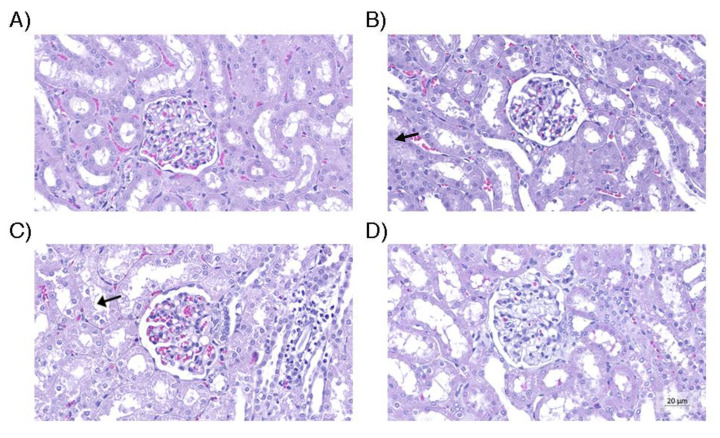
Renal tissue. (**A**) Control group containing glomerulus and normal tubular cells; (**B**) Group without use of vasopressors, containing larger tubular cells, with greater degree of edema; (**C**) Group receiving only norepinephrine, with cell enlargement and tubular cell edema (arrow), similar to group B; (**D**) Group that received the optimized strategy, in which the morphometric parameters were similar to the control group. Hematoxylin and eosin (original magnification, 200×).

**Figure 5 life-13-01439-f005:**
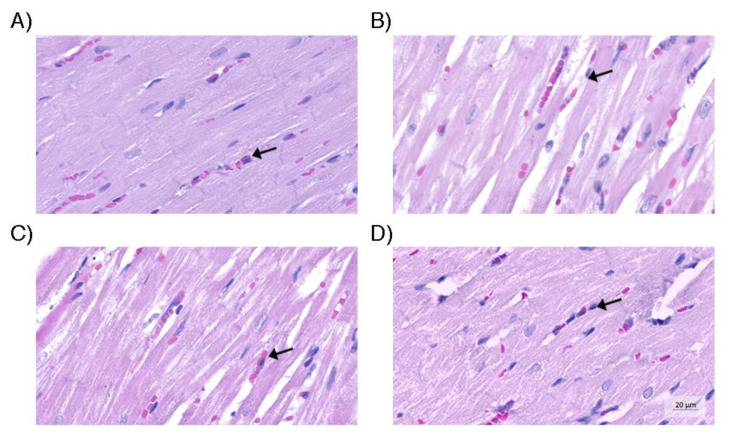
Heart tissue. (**A**) Myocardial tissue of the control group, within the parameters of normality; (**B**) Myocardium of the group without use of vasopressors with increased capillary congestion; (**C**) Myocardium of the group with norepinephrine use containing capillaries with coagulation (arrow) similar to group B; (**D**) Myocardium of the optimized strategy group containing capillaries with characteristics similar to those observed in the control group. Hematoxylin and eosin (original magnification, 200×).

**Figure 6 life-13-01439-f006:**
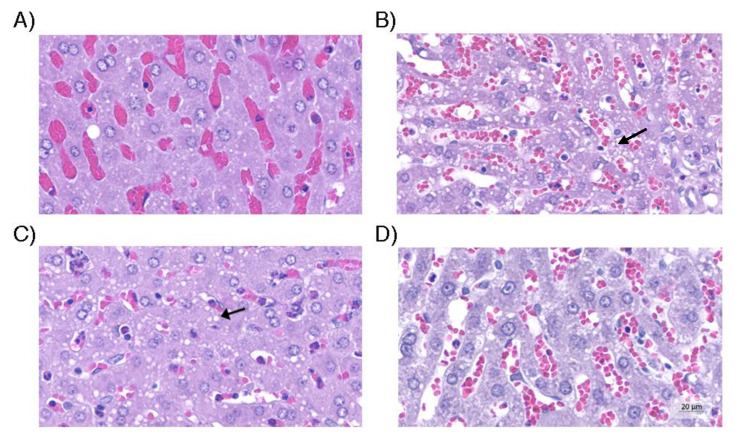
Liver tissue. (**A**) Control group with hepatocytes with no evidence of steatosis. (**B**) Hepatic tissue of the group without use of vasopressors and with microgoticular steatosis; (**C**) Group with norepinephrine use, with statistically similar degree of steatosis to the group without use of vasopressors (arrow); (**D**) Group with optimized strategy in which the degree of steatosis was minimal and close to that observed in the control group. Hematoxylin and eosin (original magnification, 200×).

## Data Availability

Publicly available datasets were analyzed in this study. This data can be found here: https://pergamum-biblioteca.pucpr.br/acervo/348764.

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
