# Peer review of "Histological, Laboratorial, and Clinical Benefits of an Optimized Maintenance Strategy of a Potential Organ Donor—A Rabbit Experimental Model"

_life, 2023, doi:10.3390/life13071439_

Round 1

Reviewer 1 Report (Previous Reviewer 1)

Dear Authors,

thank you for resubmitting your study report and making some suggested changes to it that make your work better. 

The hypothesis and background were redone and became more specific and clear. Although not particularly novel. 

Methodologically it is a well-done study with clearly stated limitations after revision. 

Results achieved are presented in abundance and in a very illustrative manner. 

The discussion part is nicely presented but again does not give an impression of added value to the medical knowledge we have now. 

Overall it is a nice small experimental study with limitations due to the size and lacking clinical parameters confirming that clinical guidelines are worth following.  

Author Response

Thanks for the fine suggestions in order to improve our research.

Reviewer 2 Report (Previous Reviewer 2)

The authors improved the manuscript considerably, but there are some aspects I would like to comment:

- In figure 6 the arrows pointing the steatosis are present only in panel C, why? In panel B the steatosis is clear.

- Once the authors mention to the reviewers their study limitations, I think it would be nice to have a limitations paragraph in the manuscript mentioning them.

- In page 18 line 353, the authors wrote “The histological parameters evaluated demonstrated hypoperfusion and early ischemia. [66–70]”. The connection made between the histological results and the ischemia is not clear. Please clarify.

- I would suggest a final text checking in order to improve any grammar, spelling and syntax mistakes.

Author Response

  1. The arrow was pointed in a visible focus of steatosis
  2. Thanks for the suggestion. Se added this explanation: "It is important to highlight that this study was performed in a small number of participants. As an experimental study, it does not intend to be conclusive, but it meant to raise hypothesis to identify biological variable that are difficult to systematically measure in humans."
  3. Ischemia was based on sinusoid congestion, hypotrophy of pericentral hepatocytes in zone 3 and also on the presence of steatotic cytoplasmic microvacuolation.

  4. We corrected some mistakes in grammar and spelling.

This manuscript is a resubmission of an earlier submission. The following is a list of the peer review reports and author responses from that submission.

Round 1

Reviewer 1 Report

Overall, this is a lovely experimental study that required a certain amount of work and I congratulate dear authors on that.

But I have some doubts about the novelty of the idea and what is the purpose of the study as described problems have solutions suggested in the existing clinical guidelines for some time now, including the use of replacement hormone therapy and vasopressors after reaching euvolemia. 

The background of the problem is well described and based on suitable references. Even though the majority of the references are more than 10 years old. Also, it would be useful if authors could identify a specific goal and purpose of the study. Maybe it could be based on evaluating the benefits of RHT on various clinical and histological criteria? 

The methodology is well-described and understandable. Statistical methods are appropriate but very small subgroups and statistics are questionable. Still, group B looks strange as there is no such strategy in clinical practice to maintain potential donors' organ perfusion using fluids only. What added value to the experiment it gives? It is difficult to understand when fluids would be stopped and how volemic status was evaluated. Could you comment on that?  Obviously, giving only NS up to 40ml/kg in such a short time would cause hypervolemia, hypernatremia, and end-organ damage. Especially, not knowing volemic status and diuresis.

Diuresis is missing as this is one of the major clinical signs in DBD maintenance allowing to evaluate hormone insufficiencies, sodium levels, hypovolemia, and also an indicator for replacement therapy. Any chance to get that in all 4 groups? 

Elementary laboratory assessment does not allow evaluation of any hyperinflammatory response to brain death, even though the authors gave the impression that it is one of the mechanisms of organ damage they intend to consider. Any chance to perform blood tests for that (e.g. IL-, TNF, endothelial injury markers )? 

I would like to ask for a comment on the choice of the histological parameters evaluated as liver steatosis is not getting changed in 4 hours.  The idea was to assess the effect of different maintenance strategies in a very short period.

The results and conclusions of the study are quite predictable according to the methodology. 

I would suggest the authors to polish the main hypothesis of this study first and work on methodology then. 

Reviewer 2 Report

This is an interesting experimental work where the authors compared the systemic and organ repercussions of brain death induction in rabbits with different maintenance regimens. The reported data are of interest. However, there remain concerns that need attention:

1-      The references mentioned are mostly old. It would be important to cite newer experimental and clinical work with important remarks regarding the brain death consequences that are not considered by the authors. For example, microcirculatory consequences and hormonal changes and treatments. The number of cited and discussed articles can also be considered low.

2-      The organs obtained include the lungs, but the authors did not mention their analysis or results. Why? Did the higher fluid infusion in some of the groups compromised the lungs? Not all the histological images presented have the scale bar and when an arrow is inserted (Figure 4C and 5C) there is a lack of the arrow to compare. “with cell enlargement and tubular cell edema (arrow), similar group B” (group B image has no arrow).

3-      The histological analysis is not well clarified. The authors mention the evaluation of glomerular space and size, and cell size. No numbers are presented, once the data is only qualitative. The same applies to the heart, muscle fibers and the average size of the vessels were evaluated. Once the main results of this manuscript were obtained by histology, the clarification and expansion of the data is necessary.

4-      The affirmation that “the fact that histological changes can be detected without significant change in laboratory tests commonly used for validation of organs for transplant is quite relevant. This fact suggests that the biochemical changes are late markers of tissue injury.” is not supported by the literature.
